# High Frequency of Viral Co-Detections in Acute Bronchiolitis

**DOI:** 10.3390/v13060990

**Published:** 2021-05-26

**Authors:** Hortense Petat, Vincent Gajdos, François Angoulvant, Pierre-Olivier Vidalain, Sandrine Corbet, Christophe Marguet, Jacques Brouard, Astrid Vabret, Meriadeg Ar Gouilh

**Affiliations:** 1Groupe de Recherche sur l’Adaptation Microbienne (GRAM 2.0), Normandie University, UNICAEN, UNIROUEN, EA2656, F-14033 Caen, France; corbet-s@chu-caen.fr (S.C.); christophe.marguet@chu-rouen.fr (C.M.); Brouard-j@chu-caen.fr (J.B.); vabret-a@chu-caen.fr (A.V.); meriadeg.legouil@normandie-univ.fr (M.A.G.); 2Laboratoire de Virologie, Centre Hospitalo-Universitaire, F-14033 Caen, France; 3Département de Pédiatrie Médicale, Centre Hospitalier Universitaire de Rouen, EA2656 Université de Normandie, UNIRouen, F-7600 Rouen, France; 4Antoine Béclère University Hospital, APHP, 92140 Clamart, France; vincent.gajdos@aphp.fr; 5Assistance Publique-Hôpitaux de Paris, Paediatric Emergency Department, Necker-Enfants Malades University Hospital, Université de Paris, 75015 Paris, France; francois.angoulvant@aphp.fr; 6INSERM, Centre de Recherche des Cordeliers, UMRS 1138, Sorbonne Université, Université de Paris, 75006 Paris, France; 7Equipe Chimie et Biologie, Modélisation et Immunologie Pour la Thérapie (CBMIT), Université Paris Descartes, CNRS UMR 8601, 75006 Paris, France; pierre-olivier.vidalain@inserm.fr; 8Equipe VIRIMI-Infections Virales, Métabolisme et Immunité, Centre International de Recherche en Infectiologie (CIRI), INSERM U1111, CNRS UMR5308, Université Lyon 1, ENS de Lyon, CEDEX 07, 69364 Lyon, France; 9Service de Pédiatrie Médicale, CHU Caen, 14000 Caen, France

**Keywords:** respiratory syncytial virus, molecular diagnostic, respiratory viruses, coronavirus, bronchiolitis

## Abstract

Over two years (2012–2014), 719 nasopharyngeal samples were collected from 6-week- to 12-month-old infants presenting at the emergency department with moderate to severe acute bronchiolitis. Viral testing was performed, and we found that 98% of samples were positive, including 90% for respiratory syncytial virus, 34% for human rhino virus, and 55% for viral co-detections, with a predominance of RSV/HRV co-infections (30%). Interestingly, we found that the risk of being infected by HRV is higher in the absence of RSV, suggesting interferences or exclusion mechanisms between these two viruses. Conversely, coronavirus infection had no impact on the likelihood of co-infection involving HRV and RSV. Bronchiolitis is the leading cause of hospitalizations in infants before 12 months of age, and many questions about its role in later chronic respiratory diseases (asthma and chronic obstructive pulmonary disease) exist. The role of virus detection and the burden of viral codetections need to be further explored, in order to understand the physiopathology of chronic respiratory diseases, a major public health issue.

## 1. Introduction

Acute bronchiolitis is the most common respiratory disease in infants under 12 months of age, and the leading cause of hospitalization in infants [1]. Respiratory syncytial virus (RSV) is the most frequently identified virus in bronchiolitis, detected in 41 to 83% of patients [2,3]. RSV is responsible for 34 million new cases of lower respiratory tract infections, and 2.4 million hospitalizations of infants all over the world, with 199,000 deaths per year, mostly in developing countries [4]. Many other viruses are found in bronchiolitis, including human rhinoviruses (HRV), *Metapneumovirus* (hMPV), coronaviruses (CoV), bocaviruses (BoV), influenza viruses, adenoviruses (ADV) and parainfluenza viruses (PIF) [2,5]. In few years, PCR methods and multivalent techniques in particular have become instrumental in detecting viruses associated with acute respiratory infections. Many studies evaluated the virus-dependent risk of outcomes, and the impact of viral etiology on severity and length of stay has been established [2,3,5,6]. Viral co-infections remain poorly investigated and available studies are scarce [2,7]. The aim of this study was to evaluate the burden of viral co-detections in infants with acute bronchiolitis.

## 2. Methods

### 2.1. Study Design, Patient Population, Sampling and Statistical Analysis

In a prospective, multicenter, double-blinded, randomized prospective study called GUERANDE, which aimed to test the efficacy of 3% hypertonic saline nebulization in acute viral bronchiolitis, respiratory samples were collected from 719 patients [8]. Twenty-four French hospital centers included patients between 15 October 2012 and 15 April 2014. Infants from 6 weeks to 12 months of age who were taken to emergency departments (ED) for a first episode of moderate to severe bronchiolitis were included. Nasopharyngeal swab samples were obtained for viral testing, and sent to the virology laboratory of the University Hospital Center of Caen (Normandy, France). The results from the two PCR techniques were compared using Kappa’s coefficient, and the risk of co-infection was calculated with Fisher’s test. Written informed consent was obtained from the parents of the patients. The Saint-Germain-en-Laye Ethics’ Committee approved the study (reference 12020).

### 2.2. Sample Processing and PCR Assay

Swab material was resuspended into 3 mL viral transport medium, divided into aliquots and stored at −80 °C until it was processed. Total nucleic acids were extracted from 300 µL of sample and eluted in 100 µL of TAE buffer using the QiaSymphony apparatus. The first aliquot was analyzed by real-time duplex RT-qPCR with Taqman hydrolysis probes in order to type RSV-A and RSV-B. The second aliquot was analyzed by multiplex RT-qPCR (Luminex NxTAG RPP) targeting 18 viruses: RSV-A and B, hMPV, influenza virus A (H1v/H3) and B, rhinovirus/enterovirus (HRV), PIF 1-4, CoV 229E, NL63, OC43, HKU1, ADV and BoV), and 3 intracellular bacteria (*Mycoplasma pneumoniae, Chlamydiae pneumoniae* and *Legionella pneumophila*).

## 3. Results

Patients’ median age was 3 months (IQR 2-5), and 719 samples were collected in the study. According to the duplex real-time RT-PCR targeting RSV, 88% (*n* = 633) of samples were positive. RSV-A- and RSV-B-positive samples account for 52% (*n* = 374) and 34% (*n* = 244), respectively. Co-detections (RSV-A and RSV-B) account for 2% (*n* = 12).

In comparison, the multiplex RT-PCR found that only 2% of samples (*n* = 12) were virus-negative (Figure 1), while 90% (*n* = 647) were RSV-positive, and 34% (*n* = 246) were HRV-positive. Co-detections (2 or more viruses) were found in 55% (*n* = 396) of samples. The most abundant co-detection was RSV-HRV (30%, *n* = 218), with or without another virus. The RSV-HRV co-detection rate without another virus was 21% (*n* = 149).

Concerning the multiplex technique, we reported 49% (*n* = 352) and 29% (*n* = 206) RSV-A-positive and RSV-B-positive samples, respectively (Figure 2). Twelve percent (*n* = 89) of samples were positive for both RSV-A and B. Among the RSV-A-positive samples, 37% were mono-infected, 20% were co-infected with HRV, 10% were co-infected with RSV-B, 12% were co-infected with another virus, 14% were co-infected with two other viruses, and 2% were co-infected with three other viruses. Among the RSV-B-positive samples, 37% were mono-infected, 13% were co-infected with HRV, 11% were co-infected with another virus, 11% were co-infected with two other viruses, and 5% were co-infected with three other viruses (Figure 2). Among the RSV-A- and B-positive samples, 52% were co-infected with another virus.

Forty-four (6.1%) discrepancies between the two PCR techniques used were found (Figure 3). On the one hand, duplex PCR found more positives than the multiplex technique, with10 positive for RSV-A and 6 for RSV-B. On the other hand, the multiplexed technique reported more positives than the duplex, with 20 positive for RSV-A, 5 for RSV-B, and 3 for RSV A + B. Kappa’s tests between the two PCR techniques show a moderate accordance for RSV-A (0.47) and a good accordance for RSV-B (0.66).

We compared the proportion of co-detections of HRV/RSV with the proportion of mono-detections of HRV: the risk of being infected by HRV is significantly higher in the absence of RSV detection (*p* < 0.001, odds ratio = 0.09, IC_95%_ 0.0537; 0.1457). The risk of being co-infected by a coronavirus and HRV or RSV is not significantly different than not being co-infected by these viruses.

## 4. Discussion

Our study enrolled 719 infants aged 6 weeks to 12 months and presenting an acute moderate to severe first episode of bronchiolitis at the emergency departments in 24 different hospitals across France. Nasopharyngeal swabs were collected and virus detection was realized by RT-qPCR.

One limitation of the study consists of the selection of the cohort’s participants and the resulting homogeneity of the cohort. Indeed, all patients were less than one year old, without bronchiolitis or wheezing history and their inclusion took place at the ED, where the most serious cases are usually observed. Moreover, there was no control group in which we could estimate the rate of asymptomatic viral infections. In 2016, Self et al. found a virus detection of 24% among asymptomatic children [9], but these controls were hospitalized for a programmed ear, nose and throat (ENT) intervention, and we know that this population is characterized by frequent infections and may not be fully representative of the general population.

We found only 2% of the samples were negative when analyzed by multiplex PCR, consistently with the literature. Mansbach et al. estimated this rate at 6% in 2012 in hospitalized patients with severe bronchiolitis in USA [5]. A low level of negative samples in children cohort is commonly seen, especially in symptomatic infants.

A high rate of viral co-detections is found in our cohort (55%), which stands between the 34% in the US EPIC study [6], which studied viral detection in children with pneumonia under 5 years of age, and 61% according to the ORAACLE Study Group (which was a Norwegian clinical cohort studying length of stay of hospitalized patients with bronchiolitis according to viruses found in PCR) [10]. These differences could be explained by the accuracy of the new molecular detection methods used here. Nevertheless, we still ignore the consequences of these co-detections, and whether this impacts the development of illnesses, particularly for future asthma or chronic obstructive pulmonary disease (COPD). HRV, which is found in 34% of cases in our cohort, is known to be associated with later asthma development [11], but no study has investigated the potential role of co-infections in the development of respiratory chronic respiratory diseases.

Several new antiviral molecules are under development [12], and their potential remains unclear, because it is not known when they should be used, and how (indications, prophylaxis). Few studies focusing on the burden of viral load exist and investigate if the viral load could be a marker to follow infections and to evaluate antiviral efficacy [13]. The severity of bronchiolitis is associated with a high RSV viral load [10], resulting in a longer length of stay and an increased frequency of oxygen and ventilation support. In the same way, severe ADV pneumonia has also been associated with a persistently high viral load [14]. This suggests that severe illnesses exhibit a high viral replication, whereas a decline of the viral load has been observed in healing patients. We need to follow up the viral load over acute clinical infections in order to provide more data. Multiplex PCR is a qualitative but not quantitative method, and viral load cannot be calculated. Studies often rely on semi-quantitative PCR methods based on the number of cycle thresholds (CT), which are sometimes correlated with the viral load in copies/mL. However, these methods are not always reproducible between different commercial machines or kits, and between different centers.

Twenty-one percent of cases are RSV/HRV co-detections, representing 38% of total co-detections. We still ignore the sequence of events leading to these co-detections. The potential exclusion or facilitating effects need to be further explored. For example, we do not know whether RSV infection may promote HRV infection, and whether co-infections occur sequentially or simultaneously. Interestingly, we found that the risk of being infected by HRV is higher in the absence of RSV, suggesting exclusion mechanisms between these two viruses. Conversely, coronavirus infection had no impact on the likelihood of co-infection involving HRV and RSV. To better characterize these potential interactions, it would be interesting to include mild bronchiolitis, which is treated by the general practitioners and represents the majority of bronchiolitis cases, and to precisely focus on the onset to sampling delay. We also propose collecting several samples for each patient, in order to study the longitudinal history of natural infections. It would be interesting to determine the viral loads in co-detections, to know which viruses are currently active and responsible for the symptoms of our patients.

## Figures and Tables

**Figure 1 viruses-13-00990-f001:**
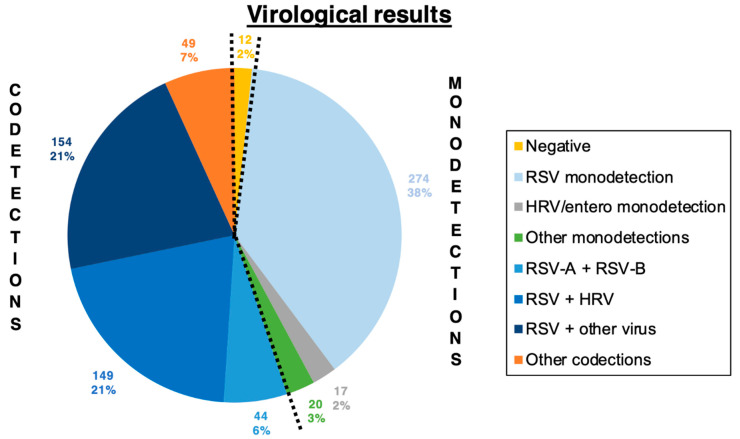
Virological results.

**Figure 2 viruses-13-00990-f002:**
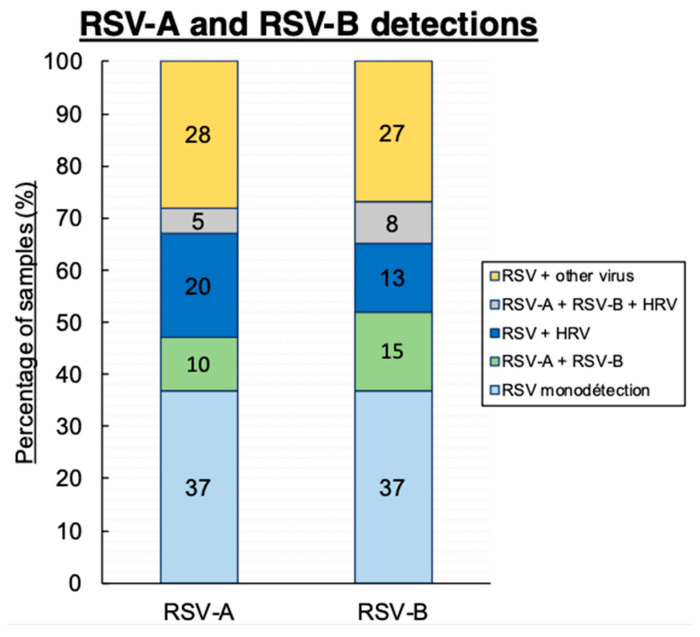
RSV-A and RSV-B detections.

**Figure 3 viruses-13-00990-f003:**
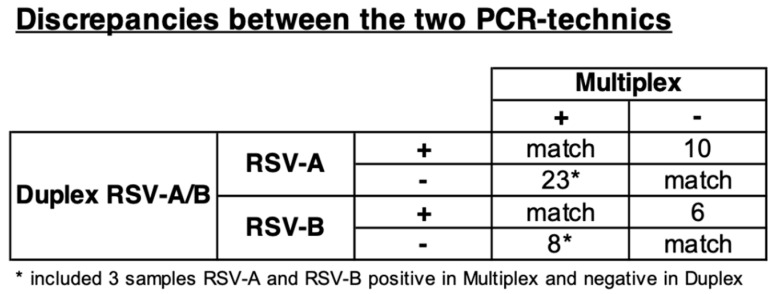
Discrepancies between the two PCR-technics.

## Data Availability

The data presented in this study are available within this article and from the GUERANDE study.

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
