# Peer review of "High Frequency of Viral Co-Detections in Acute Bronchiolitis"

_viruses, 2021, doi:10.3390/v13060990_

Round 1

Reviewer 1 Report

Acute bronchiolitis is the most common respiratory infection disease in infants under 12 months of age, according ERS guidelines. The paper by Petat et al. reports the frequency and the role of co-detections in patients hospitalized for moderate to severe bronchiolitis. The most important limitation of this study is that there is no control group and cases are a selection of a cohort’s partecipants.

MAJOR POINTS

  • Introduction: It is necessary to change the definition of bronchiolitis, following the guidelines of the ERS (under 12 months and not 2 years of age). Your patients were all under 12 months of age and their median age was 3 months. At the end of introduction, when you talk about coinfections you have to add other papers like Acute bronchiolitis: Influence of viral co-infection in infants hospitalized over 12 consecutive epidemic seasons” by L. Petrarca et al.
  • Methods: In my opinion it would be necessary to change the inclusion criteria. Why did you choose infants from 6 weeks? Can you better explain your inclusion criteria? If you consider the ERS definition of bronchiolitis, you have to include all patients under 12 months.
  • Results: Are there any differences in virus detection depending on the months of infection? You have to better explain the correlation between hRV and RSV also based on the season of infection.
  • Discussion and abstract: you have to review the discussion and abstract based on the new corrections.

MINOR POINTS

1)         Keywords: delete Coronavirus.

Author Response

  • We modified the definition of bronchiolitis following reviewer recommendation and used the definition of the ERS.
  • We understand the point of reviewer 1 on the age of inclusion. Specifically, in this study, we decided to exclude the infants before six weeks of age in order to be out of the neonatal period and avoid potential interference or biases, considering that neonates could have specific response and presentation that significantly differ from older infants.
  • We thank the reviewer 1 for this reference and we add it to the manuscript.
  • We thank the reviewer 1 for his remark on the difference of virus (species) detection according to seasonality. Indeed, it is well known that RSV and HRV have different seasonality and are preferentially detected at different month one from each other. Nevertheless, the aim of the study was not to describe this seasonality difference.
  • The aim of the study is to study co-infection frequency in regards to infections in general. In this light, we compared co-infected patients to mono-infected patients, seen as the control group in here. There is no healthy patient in this study. Moreover, we would like to underline that the cases are not a selection fromthe cohort: they represent the entire cohort.

Reviewer 2 Report

The manuscript by Petat et al deals with the thorny question of viral detection rate from respiratory samples.

There are some issues that should be addressed:

  1. The abstract does not focus on the main objective of the manuscript, that in my opinion is the comparison between two RT-PCR technique. It should be rephrased
  2. Figure 3 should be corrected, since there is a percentage of agreement among the two methods. So the +/+ and the -/- should be reported.
  3. It is strange that the authors found a 10% of RSV-A and B co-infection. How do they explain this result?
  4. In the discussion section I do not agree with the sentence: “Interestingly, we found that the risk of being infected by HRV is higher in the absence of RSV, suggesting exclusion mechanisms between these two viruses”, because RSV-HRV co-infection represent probably the most frequent co-infection of the sample.

Author Response

  1. We thank the reviewer 2 for his remarks and would like remind that the main objective of the article is to highlight the burden of viral co-detections in bronchiolitis. To PCR technics comparison is a secondary objective.
  2. We agree with the reviewer 2 suggestion and figure 3 has been amended accordingly.
  3. In our perception, finding 10% of RSV-A and RSV-B co-detections is plausible because these viruses have both the same target population (infants) and both lead to bronchiolitis. These are 2 different viruses, and there is no cross-protective immunity between them (Zhang et al, 2006).
  4. We agree with the reviewer 2 that the most frequent co-detection found is RSV-HRV. It doesn't exclude that the risk of being infected by HRV is higher in the absence of RSV, and therefore we hypothesised on the possible exclusion between these two viruses. In case of co-detection, we don't know the record of serial infections, and maybe a co-detection is possible when HRV infection is already present before, but, according to our data, not in the case of simultaneous co-infections. This is a proposal to explain the result we found.

Reviewer 3 Report

Petat et al. tested samples from a pediatric cohort with acute bronchiolitis using multiplex PCR and detected an abundance of RSV infections, either by themselves or as a co-infection with other viruses. The study is of interest to the general readership and overall well done. Some comments that could be further clarified in the text:

  • Could the authors comment on the surprisingly low number of influenza virus detections in the study?
  • The conclusion that HRV and RSV co-infections are less likely to be detected should be further clarified. Looking at figure 1, it seems like only 2% of cases were caused by HRV mono-infections, while 21% of cases were co-detections of HRV and RSV. If anything, it seems like co-infections were more commonly detected than monoinfections? Is it really fair to conclude that co-infections are unlikely to occur, or should the main conclusion rather be that HRV mono-infections are less likely to result in bronchiolitis?

Author Response

  • Finding low number of influenza cases found explanation on the fact that influenza does not infect a lot infants (Dana et al., Journ Ped Dis, 2006), but rather older children. Moreover, in the specific case of influenza disease found in infants, it rarely causes bronchiolitis.
  • We thank the reviewer 3 for his remark on the conclusion and we have amended it for clarification. Mono-infections are less likely to result in bronchiolitis could be considered as a shortcut, a bit drastic. Much important would be to estimate the burden of co-detections in infants with bronchiolits, which represent 55% of our cohort and which is a lot, comparing the level of co-detections in cohorts of older children.

Round 2

Reviewer 1 Report

The authors answered the questions and improved the paper with the reviewer's suggestions.

Thanks